# Current and Emerging Fluorescence-Guided Techniques in Glioma to Enhance Resection

**DOI:** 10.3390/cancers17162702

**Published:** 2025-08-19

**Authors:** Trang T. T. Nguyen, Hayk Mnatsakanyan, Eunhee Yi, Christian E. Badr

**Affiliations:** 1Ronald O. Perelman Department of Dermatology, New York University Grossman School of Medicine, Laura and Isaac Perlmutter Cancer Center, NYU Langone Health, New York, NY 10016, USA; 2Department of Neurology, Massachusetts General Hospital, Neuroscience Program, Harvard Medical School, Boston, MA 02129, USA; hmnatsakanyanmovsesyan@mgh.harvard.edu (H.M.); badr.christian@mgh.harvard.edu (C.E.B.); 3Department of Physiology, College of Human Medicine, Michigan State University, East Lansing, MI 48824, USA; yieunhee@msu.edu

**Keywords:** glioblastoma, 5-aminolevulinic acid (5-ALA), fluorescein sodium (FS), indocyanine green (ICG), near-infrared imaging (NIR), intraoperative imaging, neurosurgical oncology

## Abstract

Surgical removal of as much of the tumor as possible is a cornerstone of treatment for brain cancer, as it can significantly improve both survival and quality of life. In glioblastoma, complete tumor removal is rarely possible because cancer cells infiltrate healthy brain tissue and may be located near regions that control essential functions such as speech, movement, and vision. Surgeons must balance removing the tumor tissue with preserving these critical abilities. Fluorescence-guided surgery offers a valuable tool by using special dyes to make tumor cells “glow,” giving surgeons a more precise view of tumor boundaries during the operation. This review examines currently available fluorescence-guided surgery techniques, discusses their benefits and limitations, and highlights emerging innovations that hold promise for safer and more effective brain tumor removal.

## 1. Introduction

Despite extensive research, glioblastoma (GBM) poses significant therapeutic challenges due to its infiltrative nature, rapid proliferation, and resistance to conventional therapies [1,2,3]. While advancements in targeted and immunotherapy offer promise, prognosis remains poor [4,5,6]. Median overall survival under the current standard treatment regimen of maximal safe resection followed by radiotherapy and temozolomide typically remains limited to 15 to 20 months [1,7,8].

Surgical resection remains the cornerstone of treatment for GBM, the most common and aggressive primary brain tumor in adults [9,10,11]. Surgery not only extends progression-free and overall survival but also eases symptoms and improves quality of life [12,13]. Evidence consistently supports that maximal safe resection is associated with improved survival rates irrespective of patient age, functional status, or the specifics of adjunctive therapies [13]. The surgical gold standard is gross total resection (GTR) of the contrast-enhancing tumor. In select cases involving non-eloquent brain regions, surgeons may pursue supratotal resection, removing tissue beyond the radiographically defined tumor margins, to further delay recurrence [10]. However, in eloquent brain regions such as those involved in language, motor, and sensory processing surgeons must carefully balance oncologic aggressiveness with functional preservation to avoid debilitating and irreversible deficits [14,15]. This delicate interplay underscores the necessity for advanced intraoperative techniques that allow for the accurate and real-time distinction between tumor tissue and normal brain parenchyma.

To support this objective, a range of intraoperative technologies has been integrated into routine neurosurgical practice. These include image-guided neuronavigation, intraoperative ultrasonography (iUSG), intraoperative magnetic resonance imaging (iMRI), and fluorescence-guided surgery (FGS) [1,16,17,18,19,20]. Among these, FGS has garnered particular attention for its ability to enhance visualization of tumor margins using fluorescent contrast agents. Several fluorophores are currently employed in clinical practice, including 5-aminolevulinic acid (5-ALA), sodium fluorescein (SF), and indocyanine green (ICG) [21,22,23]. Each operates via distinct mechanisms: 5-ALA is metabolized by tumor cells into fluorescent protoporphyrin IX [24,25,26,27]; SF binds to serum albumin, which accumulates around the tumor due to damage to the blood–brain barrier (BBB) [24,28]; and ICG primarily binds to plasma proteins in the bloodstream [29,30,31,32].

While these agents have demonstrated meaningful improvements in the extent of resection and surgical outcomes, each has limitations. Challenges such as low tumor-to-background fluorescence ratios, limited tissue penetration, and reliance on specialized imaging equipment restrict their broader adoption [33,34,35]. Consequently, ongoing research efforts are focused on the development of next-generation fluorophores that offer enhanced specificity, deeper tissue penetration, and longer retention times [36,37,38]. Several novel agents are currently in preclinical and clinical development, aiming to expand the applicability and effectiveness of FGS in neuro-oncology [30,39].

In this review, we provide a comprehensive and up-to-date overview of FGS in the management of GBM. By evaluating the mechanisms, advantages, and limitations of existing fluorophores, alongside future innovations, we aim to underscore the evolution of FGS in the surgical management of GBM.

## 2. 5-Aminolevulinic Acid (5-ALA)

### 2.1. Mechanism

FGS using 5-ALA has emerged as a highly effective adjunct in the treatment of GBM [22,40]. Introduced into clinical practice in Germany in 1998, 5-ALA-guided resection was later validated through prospective clinical trials, which demonstrated a significant increase in the rate of complete tumor resection and an improvement in short-term progression-free survival (PFS) [41]. These promising outcomes led to the U.S. Food and Drug Administration (FDA) approval of 5-ALA in 2017 for use in patients undergoing surgery for high-grade gliomas [27,42,43,44]. Since its approval, 5-ALA has become standard practice in many leading neurosurgical centers worldwide [44,45].

Administered orally 2–4 h prior to surgery, 5-ALA is selectively taken up and metabolized by malignant glioma cells into protoporphyrin IX (PpIX), a fluorescent molecule [46]. When exposed to violet-blue light (410–420 nm) from a specially equipped surgical microscope, PpIX emits a distinct pink-red fluorescence (λ = 635 nm), enabling real-time visualization of tumor margins during resection [47]. At the infiltrative tumor margins, the fluorescence appears as a lighter pink due to lower tumor cell density and the presence of surrounding normal brain tissue [44].

At the molecular level, 5-ALA is a natural precursor in the heme biosynthesis pathway [48,49]. In healthy cells, PpIX is rapidly converted into heme by ferrochelatase, the rate-limiting enzyme in the pathway [49]. However, malignant glioma cells often exhibit decreased ferrochelatase activity, leading to the accumulation of PpIX [27,50]. This selective metabolic difference, combined with a disrupted BBB in high-grade gliomas and the loss of heme feedback regulation, promotes preferential uptake and retention of 5-ALA in tumor tissue [46] (Figure 1).

### 2.2. Limitation

Although many studies have supported the utility of 5-ALA FGS in maximizing the extent of resection for GBM, several factors may explain its limited adoption. First, the utility of 5-ALA FGS in deep-seated GBMs such as those located in the brainstem or basal ganglia remains poorly characterized [26,34]. There is currently a lack of clinical data supporting its benefits in these anatomically challenging regions, where aggressive resection carries significant risk to critical neurological functions [51]. Second, while 5-ALA preferentially accumulates in glioma cells, its fluorescence is not entirely tumor-specific. Significant levels of extracellular PpIX may be produced by non-neoplastic cells in and around the tumor margin, potentially leading to false-positive signals and inadvertent damage to normal brain parenchyma during resection [11,26]. For example, Herta et al. reported modest sensitivity (46%) and specificity (81%) of 5-ALA for detecting malignant GBM tissue, highlighting variability in its diagnostic performance [52]. Third, the practical implementation of 5-ALA is constrained by significant technical requirements [53,54]. Its effective use depends on surgical microscopes equipped with blue light-emitting sources and specialized filters to excite and detect fluorescence at precise wavelengths [55]. The combined costs of acquiring and maintaining this equipment, together with the price of 5-ALA itself, can create substantial financial challenges, particularly for resource-limited institutions. Finally, safety concerns also persist, particularly regarding potential phototoxicity and the uncertain implications of prolonged exposure to fluorescent light [56].

### 2.3. Clinical Applications

The phase 1/2 clinical trial NCT01128218 evaluated escalating doses of 5-ALA, ranging from 10 to 50 mg/kg, in patients with high-grade gliomas to improve intraoperative tumor visualization [57,58,59,60]. The study utilized 5-ALA to differentiate malignant glioma tissue from normal brain tissue (NCT00241670) during surgery, with fluorescence signals compared to biopsy-confirmed pathology. The study concluded that 5-ALA is safe and effective for FGS in malignant gliomas, with a recommended dose of 40 mg/kg established in phase 2. These findings support the clinical utility of 5-ALA in enhancing the precision and completeness of brain tumor resections. In another trial, NCT00752323, researchers compared qualitative visual fluorescence and quantitative PpIX levels with histopathological findings to determine the optimal dose of 5-ALA for tumor visualization. The study found that doses of either 10 mg/kg or 20 mg/kg offered the best balance between sensitivity and specificity for accurately identifying tumor tissue during surgery.

One study investigated whether residual fluorescent tissue observed during FGS using 5-ALA is associated with prognosis in GBM patients who showed complete resection of enhancing tumors on early postoperative MRI [61]. Among 52 newly diagnosed GBM patients, those with no residual fluorescence had a significantly longer median overall survival (27.0 months) compared to those with residual fluorescence (17.5 months), a difference that remained significant after adjusting for clinical variables [61]. However, the rate of neurological complications did not differ significantly between the groups. These findings suggest that the presence of residual fluorescent tissue, even when MRI shows complete resection, may indicate a worse prognosis in GBM [61].

A phase 3 randomized clinical trial evaluated the efficacy of FGS with 5-ALA (20 mg/kg) in improving the extent of malignant glioma resection [59]. 322 patients (aged 23–73 years) with suspected malignant gliomas were assigned either to 5-ALA-guided resection or conventional white-light surgery. The primary goals were to measure the completeness of tumor removal on early postoperative MRI and PFS at 6 months [59,62]. Results from an interim analysis of 270 patients showed that complete resection of contrast-enhancing tumor occurred significantly more often in the 5-ALA group (65%) compared to the white-light group (36%) [59]. Additionally, 6-month PFS was higher in the 5-ALA group (41.0% vs. 21.1%) [59]. There was no significant difference between groups in the frequency of severe adverse events or complications within 7 days after surgery (NCT00241670) [59].

In a similar approach to the study by Stummer et al., a phase 3 randomized, open-label, multicenter clinical trial in China (NCT06160492) is evaluating the efficacy of 5-ALA HCl fluorescence-guided microsurgery versus conventional white light microsurgery in patients with WHO grade 3/4 malignant gliomas [59]. Participants are randomized 1:1 to receive either oral 5-ALA (20 mg/kg) followed by fluorescence-guided tumor resection or standard white light-guided resection. The trial is currently ongoing and still recruiting patients, with no results available at this time.

More recently, the RESECT trial, a multicenter phase 3 randomized controlled study conducted across 21 French neurosurgical centers, compared 5-ALA (20 mg/kg) FGS with conventional white-light microsurgery in patients with GBM receiving standard-of-care therapy [40]. Among 136 evaluable patients, the rate of GTR was significantly higher in the 5-ALA group (79.1%) than in the white-light group (47.8%) (NCT01811121) [40]. This benefit remained significant after adjustment for patient age, tumor location, and performance status. Importantly, postoperative neurological outcomes and functional status were comparable between groups, and adverse events associated with 5-ALA were rare and mild [40]. Although PFS and overall survival (OS) did not differ significantly between arms, GTR itself was an independent predictor of improved PFS, reinforcing the clinical value of 5-ALA-guided resection (Table 1).

## 3. Fluorescein Sodium (FS)

### 3.1. Mechanism

FS is a green-fluorescent dye with a well-established safety profile and a long history of clinical use since the 1940s [23]. It has been applied in a range of medical settings, including neurosurgery. Its utility in GBM surgery stems from its ability to highlight areas of BBB disruption, a hallmark of this aggressive and highly vascularized tumor type [47,63,64]. In contrast to 5-ALA, which relies on tumor-specific metabolic processes, FS operates via a passive mechanism, accumulating in the extracellular space of regions where the BBB is disrupted [47,65].

Following intravenous administration (5 mg/kg body weight), FS circulates systemically and rapidly accumulates in areas of BBB breakdown [47]. Under specific excitation wavelengths (typically 465–490 nm), it emits a bright green fluorescence [66]. When used with a specialized surgical microscope equipped with a yellow 560 nm filter, this fluorescence provides high contrast between tumor and normal brain tissue [44].

Notably, fluorescein also highlights infiltrative tumor margins that may not be visible under standard white-light microscopy or even on preoperative MRI, aligning well with the clinical behavior and imaging characteristics of GBM [44]. Its rapid pharmacokinetics, low cost, and wide availability make FS an appealing alternative or adjunct to agents like 5-ALA, especially in resource-limited settings [43] (Figure 2).

### 3.2. Limitation

FS is a low-cost fluorescent agent used in various medical fields, particularly ophthalmology, but its application in brain tumor surgery remains limited due to several challenges [35,67,68]. Although FS has been used since 1947 and effectively highlights regions with a disrupted BBB, it lacks tumor specificity. Its mechanism, based on vascular permeability, can also label non-tumor areas such as necrosis or inflammation [69,70,71]. Although FS has been widely adopted in Europe and has FDA approval for neurosurgical use in the U.S., concerns over dosing, side effects, and optimal timing remain [72].

### 3.3. Clinical Applications

One study conducted by the Department of Neurological Surgery at Columbia University evaluates the safety and effectiveness of using FS as a visual aid during GBM surgery to improve the extent of tumor resection. Thirty-two patients received intravenous fluorescein (3 mg/kg), which was visualized intraoperatively using a specialized surgical microscope [23]. Biopsies were taken from both contrast-enhancing (CE) and non-contrast-enhancing (NCE) tumor regions to assess fluorescence intensity and correlate it with histopathological changes. Bright fluorescence was consistently observed in CE and parts of NCE regions, effectively guiding resection. GTR was achieved at 84%. Fluorescence correlated strongly with tumor pathology in both regions, showing over 96% positive predictive value in NCE areas [23]. The study concludes that fluorescein is a safe and effective tool for guiding GBM resection, including infiltrative tumor margins beyond the CE zone [23].

A phase 3 trial (NCT03291977) conducted at Rennes University Hospital evaluated the effectiveness of fluorescein-guided resection compared to standard white-light surgery in 51 adult patients with GBM, aiming to assess the clinical relevance of this technique in tumor removal. Fluorescéine Sodique Faure was administered intravenously at a dose of 3 mg/kg, diluted in 50 mL of physiological saline, over a 10 min period during anesthesia induction.

Another study, conducted at Ibrahim Cardiac Hospital from September 2021 to November 2023, evaluated the safety and efficacy of FS (5 mg/kg) as a surgical aid in GBM resection [28]. The study included 12 patients who received intravenous FS 30 min before surgery. GTR was achieved in 66.6% of cases, with no severe adverse effects observed apart from one manageable postoperative seizure. Minor, transient side effects included yellow discoloration of skin, sclera, and urine. All patients maintained normal liver and kidney function post-surgery. With a median follow-up of 13.4 months, the findings support FS as safe, effective, and cost-efficient [28]. A similar finding was reported by Hohne et al., where FS (5 mg/kg) enabled clear visualization of tumor tissue, resulting in GTR in 84% of 106 patients with recurrent GBM, without any reported adverse effects [66].

In another study evaluating FS (3–5 mg/kg) safety and efficacy in GBM resection, 32 patients received intravenous fluorescein prior to surgery [73]. GTR was achieved in 84% of all cases. Median OS of the FGS group was 4.2 months longer than the non-FGS group despite no statistical difference (18.2 months vs. 14.0 months, HR 0.63, 95% CI 0.36–1.11, *p* = 0.112) [73]. Sodium fluorescein-guided surgery for high-grade gliomas in eloquent and deep-seated brain regions enables more extensive resection while preserving neurologic function and improving patient survival [73].

In phase 2 trial NCT02691923, FS is evaluated as an intraoperative imaging biomarker during surgery for patients with presumed high- or low-grade gliomas [74,75,76]. Participants are randomized to receive either intravenous fluorescein alone (5 mg/kg, administered approximately 30 min before tumor resection) or a combination of fluorescein and oral 5-ALA (20 mg/kg) [75]. Randomization is stratified by tumor grade, with a 2:1 ratio for high-grade glioma patients and 1:1 for low-grade glioma patients. Pre- and post-operative imaging used for analysis is obtained as part of routine clinical care. Adverse events are monitored through standard clinical follow-up, with additional liver function monitoring in patients receiving 5-ALA.

A phase 2 clinical trial, NCT04597801, was conducted to compare the diagnostic accuracy of two intraoperative techniques for brain tissue evaluation: FS (5 mg/kg administered intravenously 20–40 min prior to tumor resection) combined with in vivo confocal microscopy and conventional frozen section analysis. The study also evaluated safety by monitoring adverse events and comparing the time required for each technique during surgery (Table 1).

The integration of FS with in vivo microscopy enables surgeons to directly visualize fluorescent malignant tissue, allowing real-time assessment of tumor cells and immediate, precise adjustments to their surgical approach. However, most clinical trials involving FS have included relatively small patient cohorts, which may limit the generalizability and statistical strength of the results. Therefore, larger-scale studies are necessary to confirm the efficacy and safety of FS.

## 4. Indocyanine Green (ICG)

### 4.1. Mechanism

ICG is an amphiphilic near-infrared (NIR) fluorescent dye with a long-standing clinical track record [77,78]. First approved by the FDA in 1959 for liver function testing, ICG was subsequently adopted in ophthalmology for visualizing retinal vasculature due to its strong binding to plasma proteins and its localization within the bloodstream [79]. Its clinical utility has expanded considerably to include cardiovascular, hepatic, and particularly neurosurgical imaging [21,39,80]. ICG was administered to patients via intravenous injection [81]. It binds primarily to plasma proteins such as α1-lipoproteins and stays within the blood vessels under normal integrity of blood vessels [32]. ICG is not metabolized and is exclusively cleared by the liver, with a plasma half-life of approximately 3–4 min.

For ICG video angiography, the typical dosage ranges from 0.2 to 0.5 mg/kg, with a recommended maximum daily limit of 5 mg/kg [32]. During surgery, an NIR laser excites the dye, and the resulting fluorescence is captured by a filtered digital video camera, enabling real-time visualization of vascular structures [36]. ICG is widely used in GBM surgeries due to its ability to accumulate in tumor tissue where the BBB is disrupted [82,83]. When activated by NIR light, ICG enables real-time imaging with deeper tissue penetration and minimal background interference [84]. (Figure 3).

NIR fluorescence imaging within the NIR-I window (700–900 nm) is well-established in clinical practice for diagnostic imaging and image-guided surgery, largely enabled by FDA-approved dyes and the development of new targeted probes [85]. Recently, there has been growing interest in imaging within the NIR-II window (1000–1700 nm), which offers several advantages over NIR-I, including reduced tissue autofluorescence, lower photon scattering, and greater tissue penetration [36,85]. These attributes enhance spatial resolution and image contrast, making NIR-II imaging a promising frontier for future clinical applications. ICG enables tissue penetration of up to 15 mm but requires specialized imaging systems for detection [86]. In a cohort of 23 patients undergoing fluorescence-guided liver tumor resection with indocyanine green, NIR-II imaging demonstrated superior performance, with higher tumor-detection sensitivity (100% vs. 90.6%) and an improved detection rate (56.41% vs. 46.15%) compared to NIR-I [85] (Figure 3).

### 4.2. Limitation

ICG is primarily eliminated via biliary excretion and is generally well-tolerated, with no major side effects reported. Despite its safety profile, the clinical utility of ICG is limited by several factors. Its weak signal intensity and strong binding to plasma proteins can hinder effective accumulation at tumor sites [21,84]. Additionally, its low specificity may produce false-positive signals from necrotic or inflamed tissue [84,87]. However, ICG remains a valuable tool in both vascular and oncological neurosurgery due to its established safety and versatility.

ICG has regained attention for its strong tail emission properties in the NIR-II biological window. To enhance tumor-specific targeting, recent studies have focused on conjugating ICG with NIR fluorescent dyes attached to tumor-targeting agents, such as those directed at the epidermal growth factor receptor (EGFR), which is overexpressed in 50–70% of GBM [21,88,89,90]. Further details are provided in the following section.

## 5. Antibody-Based Probes for Fluorescence Imaging

### 5.1. Cetuximab-IRDye 800

EGFR is a transmembrane protein that plays a key role in cell growth and proliferation [91]. In GBM, EGFR is either overexpressed or mutated in approximately 40% to 70% of cases [92]. This aberrant expression not only contributes to tumor progression and resistance to therapy but also provides a valuable molecular target for diagnostic and therapeutic strategies [93,94].

Recent advances in cancer imaging have utilized NIR dyes conjugated to monoclonal antibodies that selectively bind to tumor-associated antigens. For GBM and other cancer types, this approach has shown promise in enhancing tumor visualization during surgical resection. Among these, cetuximab, a chimeric monoclonal antibody targeting EGFR, has been successfully conjugated with IRDye800 to create a targeted imaging agent known as cetuximab-IRDye800 [89,90,95]. Preclinical studies using orthotopic animal models of GBM have demonstrated that cetuximab-IRDye800 effectively localizes tumor tissue, enabling real-time visualization of tumor margins under an NIR imaging system [94,95,96].

The clinical translation of this technology began with the first-in-human study published in 2018, which demonstrated that cetuximab-IRDye800 could enhance the extent of primary tumor resection and facilitate the intraoperative identification of residual tumor tissue [89]. Although these preliminary findings are promising, the trial’s small sample size of only two patients limits broader clinical validation [89]. Importantly, no severe adverse effects have been reported following cetuximab-IRDye800 administration, indicating a favorable safety profile. Therefore, further studies involving larger and more diverse patient populations are essential to confirm its safety, efficacy, and reproducibility.

### 5.2. Bevacizumab-IRDye800CW

Meningiomas are the most common primary brain tumors in adults, accounting for approximately one-third of all intracranial tumors [97,98]. These tumors often cause symptoms by compressing adjacent brain tissue. Surgical resection remains the primary treatment strategy, with GTR being essential to reduce the risk of recurrence. However, subtotal resection when tumors are located near critical neural or vascular structures is associated with a recurrence due to undetected tumor remnants left behind during surgery. To enhance tumor visualization and improve surgical outcomes, intraoperative detection using fluorescent tracers that target tumor-specific biomarkers has emerged as a promising approach. Vascular endothelial growth factor α (VEGFα) is a well-established biomarker in meningiomas and can be selectively targeted by bevacizumab conjugated to the IRDye800CW, forming the tracer bevacizumab-IRDye800CW [99,100]. This tracer has demonstrated clinical utility in detecting VEGF-overexpressing tumors in various cancer types [90,101].

The phase I LUMINA trial was conducted to evaluate the safety and feasibility of bevacizumab-IRDye800CW for intraoperative imaging in meningioma surgery and to determine the optimal imaging dose [102]. The study found that the tracer was safe and well tolerated, with 10 mg providing a sufficient tumor-to-background fluorescence ratio. Based on these promising results, a larger phase 2/3 trial is warranted to further assess the clinical benefits of fluorescence-guided surgery using bevacizumab-IRDye800CW in patients with meningioma [102].

### 5.3. MCT4-ICG-NIR-II

Monocarboxylate transporter 4 (MCT4), which exports lactate from cells and is overexpressed in various cancers under hypoxic conditions, has emerged as a valuable target due to its association with poor prognosis and absence in normal brain tissue [103,104]. A recent study designed a probe combining an MCT4-specific antibody with indocyanine green for NIR-II imaging and photothermal therapy. The probe enabled accurate tumor targeting, achieved high signal-to-background ratios, penetrated the BBB, and elevated tumor temperatures sufficiently to ablate cancer cells. This approach significantly reduced tumor burden and prolonged survival in mice, with no observed toxicity to vital organs, underscoring its potential to improve glioma surgery and treatment outcomes [105].

### 5.4. Miltuximab-NIR

There is compelling evidence that Glypican-1 (GPC-1), a cell surface antigen overexpressed in various solid tumors but largely absent in normal tissues, plays a key role in the progression of cancers such as prostate, pancreatic, bladder, gastroesophageal, ovarian, and GBM [106,107,108,109,110]. As such, a theranostic agent targeting GPC-1 holds significant promise for clinical application. One study demonstrates the potential of Miltuximab^®^-IR800 fluorescent antibody targeting GPC-1 for specific and effective molecular imaging of GBM [111]. The antibody–dye conjugate showed strong tumor accumulation and retention in GBM xenografts, validating its use for FGS. Unlike traditional dyes such as fluorescein and 5-ALA, which suffer from low specificity and high background autofluorescence, Miltuximab^®^-IR800 benefits from NIR imaging’s deeper tissue penetration and reduced background signal [111]. The conjugate demonstrated high binding specificity, a stable signal over 10 days, and no toxicity in preclinical models [111].

## 6. Peptide-Based Probes for Fluorescence Imaging

BLZ-100 (Tozuleristide) is a tumor-targeting imaging agent composed of chlorotoxin (CTX) conjugated to ICG [47]. CTX is a 36-amino acid peptide originally derived from scorpion venom and has shown minimal toxicity in human trials [112]. It selectively binds to tumor cells through interactions with Annexin A2 and matrix metalloproteinase-2 (MMP-2) [113,114]. When conjugated with ICG, the resulting compound is known as BLZ-100 or “Tumor Paint”. This technique is applicable to both high- and low-grade gliomas. In preclinical and early clinical studies, BLZ-100 demonstrated strong tumor localization, with higher fluorescence intensity in high-grade gliomas compared to low-grade tumors [115].

A phase 1 clinical trial investigated the safety and imaging capabilities of BLZ-100 in adult patients with newly diagnosed or recurrent glioma [116]. Participants received a single intravenous dose ranging from 3 to 30 mg, administered between 3 and 29 h before surgery. The agent was well tolerated across all dose levels, with no serious drug-related adverse events observed. These findings support the safety of BLZ-100 at doses up to 30 mg [116]. Based on pharmacokinetic modeling, an optimal dose of 15 mg/m^2^ administered intravenously 24 h before surgery was identified for pediatric populations. To further evaluate efficacy, a randomized, blinded phase 2/3 trial (NCT03579602) was initiated to enroll 114 patients. The primary objective is to assess the sensitivity and specificity of BLZ-100 for intraoperative tumor visualization. Secondary objectives include evaluating the performance of imaging systems and determining the agent’s impact on the extent of resection while preserving surrounding healthy brain tissue. Although earlier studies suggest that BLZ-100 is safe and may enable tumor-specific fluorescence, its clinical efficacy and effect on surgical outcomes remain unconfirmed. To date, no official conclusions have been released from the pivotal phase 2/3 trial (NCT03579602), and full results from this larger trial are awaited (Table 1).

## 7. Conclusions

The Korean Society for Neuro-Oncology (KSNO) and the National Comprehensive Cancer Network (NCCN) recommend using FGS with 5-ALA to help remove GBM tumors during surgery [117,118,119]. 5-ALA FGS is commonly used to achieve complete tumor removal in high-grade gliomas, but it has some important limitations. In low-grade gliomas, the fluorescence signal is often weak or missing because not enough of the substance builds up in the tumor. Also, 5-ALA relies on blue light, which does not penetrate deeply into tissue and can cause damage from light exposure. Similarly, FS has problems with low accuracy and limited imaging depth, making it less useful in complicated brain surgeries (see Table 2).

In contrast, NIR fluorescence imaging presents several advantages, including deeper tissue penetration, reduced background autofluorescence, and real-time intraoperative visualization. These benefits enable more precise delineation of tumor margins, even in deep-seated or infiltrative lesions, thereby improving surgical precision and patient outcomes. In contrast to 5-ALA, which undergoes metabolic processing in the body, NIR imaging agents remain chemically unchanged and do not break down by metabolic enzymes. Instead, these NIR agents are primarily removed from the bloodstream through the liver’s filtration and excretion processes. Because they are rapidly cleared by the liver, they have a relatively short presence in the blood, with a plasma half-life of approximately 3–4 min. In addition to these pharmacokinetic advantages, NIR imaging may provide a more cost-effective alternative to 5-ALA, reducing both agent expenses and the complexity of handling and training, while still maintaining high imaging performance [37]. Given these benefits, NIR fluorescence imaging emerges as a promising alternative to traditional fluorophores like 5-ALA and FS, with the potential to become a new standard in glioma surgery (Table 2).

However, ICG’s strong affinity for plasma proteins can limit its accumulation at tumor sites, thereby reducing its specificity. To address this limitation, recent advancements in cancer imaging have focused on conjugating NIR dyes to monoclonal antibodies that selectively target tumor-associated antigens such as Cetuximab-IRDye 800, Bevacizumab-IRDye800CW, MCT4-ICG-NIR-II, and Miltuximab-NIR, as well as peptide-based probes such as BLZ-100. These targeted imaging agents have shown significant promise in improving intraoperative tumor visualization and guiding more effective resections. Fluorescently labeled monoclonal antibodies play a crucial role in enabling more personalized surgical strategies across a wide range of solid tumors. Despite these advantages, several significant challenges remain. One major obstacle is the need for preoperative administration several days before surgery, which often requires patients to make additional hospital visits [120]. Furthermore, rapid fluorescence quenching can occur, potentially compromising imaging accuracy during the procedure [120]. Additionally, many NIR dye–antibody conjugates are still in preclinical development or early-phase clinical trials involving limited patient numbers, preventing definitive conclusions about their clinical utility. Given these considerations, patients should work closely with their medical team to determine the most suitable fluorescence-guided method. Individual factors, such as medical history and potential dye sensitivities, should be considered, as certain patients may develop adverse events like hypotension with 5-ALA or iodine hypersensitivity with ICG.

The impact of NIR imaging could be further enhanced by integrating it with complementary technologies such as handheld probes and intraoperative MRI (iMRI). These innovations improve detection sensitivity, compensate for intraoperative brain shift, and provide additional anatomical and functional information [20,121,122,123]. For example, the randomized clinical trial NCT01479686, which involved 188 patients with high-grade gliomas, demonstrated that iMRI significantly increased the extent of tumor resection, achieving a GTR rate of 83.85% compared to 50% with conventional neuronavigation [20]. Supporting this, a study assessing the combined use of iMRI and 5-ALA revealed that for tumors near eloquent brain regions, 5-ALA alone resulted in a 72% GTR rate, which increased to 100% when iMRI was added. Likewise, in tumors located within eloquent areas, GTR improved from 58% with 5-ALA alone to 71% with iMRI integration. Collectively, these findings highlight the potential of combining NIR fluorescence imaging with iMRI to enhance tumor resection and ultimately improve patient outcomes [124].

Furthermore, the incorporation of machine learning (ML) algorithms can enhance specificity and predict tumor invasion patterns, contributing to the advancement of precision neurosurgery and image-guided therapy for GBM [125]. One study investigated a fluorescence-based ML approach for intraoperative brain tumor classification by training and testing various models on fluorescence spectral data processed in Python. To optimize performance, the researchers applied strategies such as adjusting the number of pixels per sample, balancing class representation, and employing 5-fold cross-validation for model tuning. Despite challenges related to noisy training data, tumor heterogeneity, and subjective margin labeling, the models achieved strong accuracy up to 85.7% and demonstrated notable success in predicting IDH mutation status with 93% accuracy. Future enhancements, including the adoption of convolutional neural networks, improved spectral unmixing, and standardized labeling protocols, could further elevate model performance. Overall, this work underscores the growing feasibility and potential of ML-integrated, fluorescence-guided systems to enhance intraoperative decision-making and surgical outcomes in neurosurgery.

**Table 2 cancers-17-02702-t002:** Comparison of fluorescent agents used in glioma surgery [37,126,127].

Agent	5-ALA	FS	ICG-NIR
Mechanism	Bind to PpIX in tumor	Bind to BBB disruption	Bind to plasma protein
Excitation/Emission	410–420/635 nm	465–490/520–530 nm	NIR I: 780–805/820–835 nm NIR II: 900–1100/1000–1700 nm
Depth Penetration	Shallow	Shallow	Deeper (via NIR)
Limitations	Phototoxicity, equipment required	False positives, non-specific	False positives, non-specific
Cost	High	Low	Moderate

## Figures and Tables

**Figure 1 cancers-17-02702-f001:**
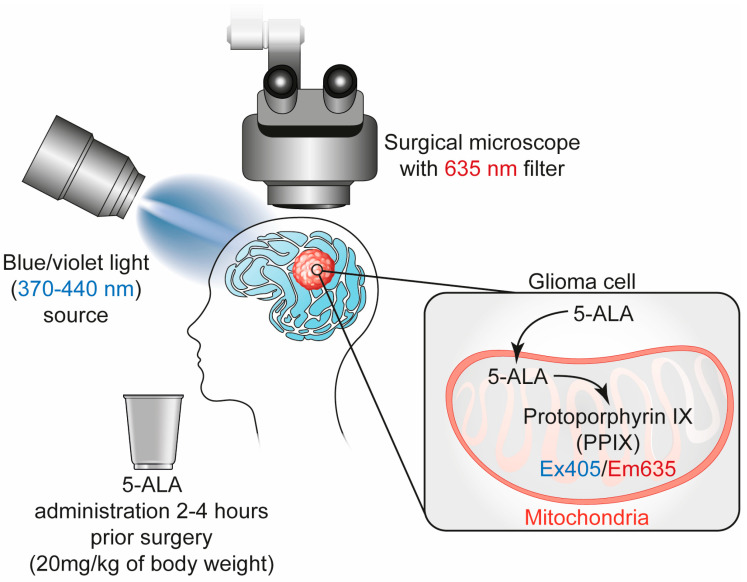
Real-time visualization through 5-ALA fluorescence-guided surgery. 5-ALA is administered orally 2–4 h before surgery, where it is selectively taken up by malignant glioma cells and converted into the fluorescent compound protoporphyrin IX (PpIX). Due to reduced ferrochelatase activity in glioma cells, PpIX accumulates to higher levels. When illuminated with violet-blue light (370–440 nm) from a specialized surgical microscope, PpIX emits a pink-red fluorescence (λ = 635 nm), enabling real-time visualization of tumor margins.

**Figure 2 cancers-17-02702-f002:**
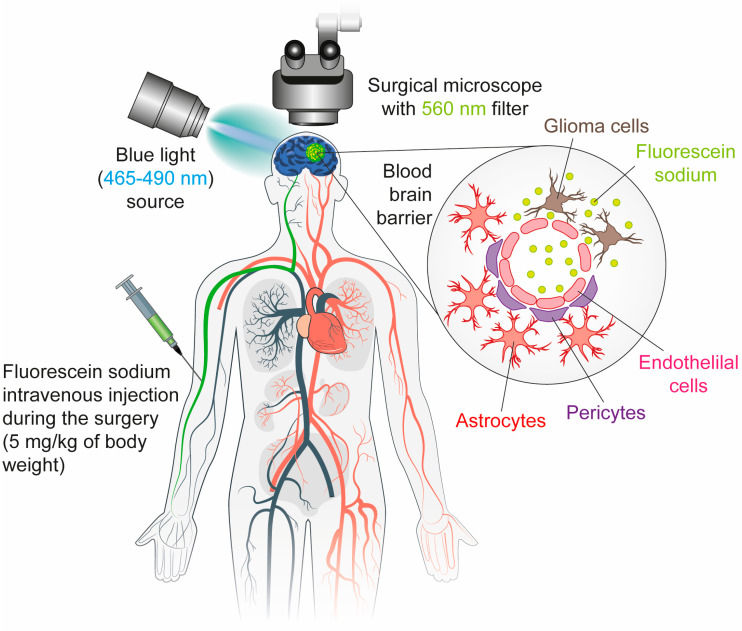
Fluorescein sodium enhances tumor visualization in fluorescence-guided surgery via BBB disruption. FS (5 mg/kg body weight) is administered intravenously immediately after the induction of general anesthesia, allowing systemic circulation and rapid accumulation in areas of BBB disruption. When exposed to blue light (465–490 nm), it emits a bright green fluorescence. Viewed through a surgical microscope equipped with a 560 nm yellow filter, this fluorescence provides high contrast between tumor tissue and the surrounding normal brain.

**Figure 3 cancers-17-02702-f003:**
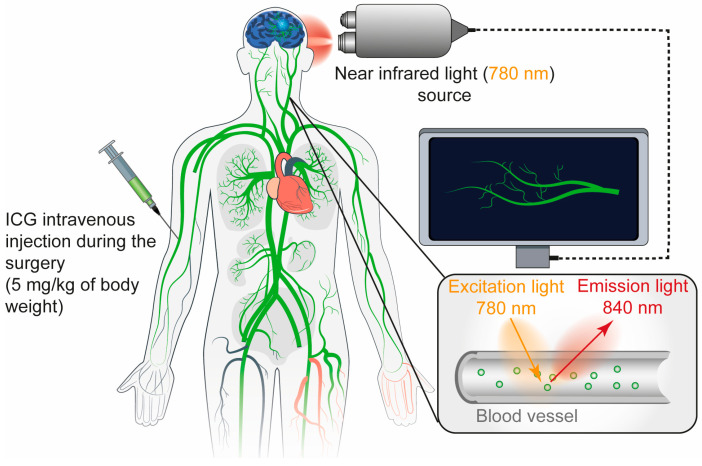
Real-time vascular imaging in fluorescence-guided surgery using indocyanine green and near-infrared light. Indocyanine green is administered intravenously up to 5 mg/kg, where it binds predominantly to plasma proteins like α1-lipoproteins and remains confined to the vasculature unless the BBB is disrupted. During the procedure, NIR light excites the dye, and the resulting fluorescence is captured by a filtered digital camera, enabling real-time visualization of vascular structures.

**Table 1 cancers-17-02702-t001:** Current and completed clinical trials of 5-ALA, FS, and NIR in GBM from ClinicalTrials.gov.

Clinical Trial Identifier/Therapy	Study Name	Phase/Recruitment Status	Key Findings/Conclusions
NCT01128218/5-ALA	A study of the specificity and sensitivity of 5-ALA fluorescence in malignant brain tumors	Phase 1,2/ Completed	High-dose oral 5-ALA (>40 mg/kg) is safe and effective for intraoperative tumor detection. However, it does not significantly reduce false-negative observations compared to standard dosing [60].
NCT00241670/5-ALA	Fluorescence-guided resection of malignant gliomas with 5-ALA	Phase 3/ Completed	5-ALA facilitates more complete resection of contrast-enhancing tumors, resulting in improved progression-free survival [59].
NCT00752323/5-ALA	Imaging procedure using ALA in finding residual tumor in grade IV malignant astrocytoma	Phase 2/ Completed	No formal conclusions or outcome data are available.
NCT06160492/5-ALA	Phase III clinical trial evaluating the resection efficacy of 5-Aminolevulinic acid hydrochloride (5-ALA HCl) fluorescence-guided microsurgery versus conventional white light microsurgery in patients with malignant glioma (WHO Grade 3/4)	Phase 3/ Recruiting	No formal conclusions or outcome data are available.
NCT01811121/5-ALA	Medico-economic evaluation of surgery guided by fluorescence for the optimization of resection of glioblastoma (RESECT)	Unknown Status	5-ALA–guided fluorescence surgery is a safe, easy-to-use, cost-effective, and time-efficient technique that enhances the extent of tumor resection [40].
NCT03291977/FS	Interest of fluorescein in fluorescence-guided resection of gliomas (FLEGME)	Phase 3/ Completed	The fluorescein-guided technique significantly improved the extent of tumor resection. However, data on progression-free survival, overall survival, and adverse events were not publicly reported.
NCT02691923/FS and 5-ALA	Diagnostic performance of fluorescein as an intraoperative brain tumor biomarker	Phase 2/ Recruiting	No formal conclusions or outcome data are available.
NCT04597801/FS	Comparison of fluorescein-intra-vital microscopy versus conventional frozen section diagnosis for intraoperative histopathological evaluation (INVIVO)	Phase 2/ Completed	No formal conclusions or outcome data are available.
NCT03579602/BLZ-100	Study of Tozuleristide and the canvas imaging system in pediatric subjects with CNS tumors undergoing surgery	Phase 2, 3/ Completed	No formal conclusions or outcome data are available.

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
