# Peer review of "Current and Emerging Fluorescence-Guided Techniques in Glioma to Enhance Resection"

_cancers, 2025, doi:10.3390/cancers17162702_

Round 1
Reviewer 1 Report
Comments and Suggestions for Authors
This is a well-structured and comprehensive review that provides an updated overview of current and emerging fluorescence-guided techniques (FGS) in glioma surgery. The topic is highly relevant, and the manuscript presents a clear discussion of clinical applications, limitations, and future directions. However, some aspects require clarification or further development to enhance scientific rigor and clinical value.
Majot revisions:
- While the manuscript mentions multiple ongoing trials and applications of FGS, it would be useful to include more discussion on current clinical guidelines or consensus statements regarding the use of 5-ALA, fluorescein, or ICG in glioma surgery (e.g., EANO, AANS, NCCN, etc.).
-
The level of evidence from included clinical trials is not always critically discussed. For instance:
-
Some trials are single-center and non-randomized.
-
The sample size for new agents like cetuximab-IRDye800 is very small.
Consider summarizing limitations of the evidence (e.g., Table format or in the conclusion)
-
- The section on NIR-II and antibody/peptide-based probes is promising but mostly preclinical. It would be beneficial to add a paragraph discussing barriers to clinical translation, such as regulatory hurdles, production costs, or imaging system availability.
- English revisions
Major english revisions
Author Response
Thank you for taking the time to review our manuscript and for providing thoughtful and constructive feedback. We sincerely appreciate your comments, which have helped us improve the clarity and quality of our work.
Reviewer 1:
This is a well-structured and comprehensive review that provides an updated overview of current and emerging fluorescence-guided techniques (FGS) in glioma surgery. The topic is highly relevant, and the manuscript presents a clear discussion of clinical applications, limitations, and future directions. However, some aspects require clarification or further development to enhance scientific rigor and clinical value.
Major revisions:
While the manuscript mentions multiple ongoing trials and applications of FGS, it would be useful to include more discussion on current clinical guidelines or consensus statements regarding the use of 5-ALA, fluorescein, or ICG in glioma surgery (e.g., EANO, AANS, NCCN, etc.).
Response: We have incorporated relevant recommendations and consensus statements from the European Association of Neuro-Oncology (EANO), Comprehensive Cancer Network (NCCN) and Korean Society for Neuro-Oncology (KSNO) in the Conclusion section (page 12, lines 422-426).
“While the Korean Society for Neuro-Oncology (KSNO) and National Comprehensive Cancer Network (NCCN) guidelines include FGS with 5-ALA as part of intraoperative strategies to enhance GBM resection, the European Association of Neuro-Oncology (EANO) guidelines emphasize structural resection and molecular diagnostics, and do not currently endorse fluorescence-based visualization due to limited supporting evidence.”
The level of evidence from included clinical trials is not always critically discussed. For instance:
- Some trials are single-center and non-randomized.
- The sample size for new agents like cetuximab-IRDye800 is very small.
Consider summarizing limitations of the evidence (e.g., Table format or in the conclusion)
The section on NIR-II and antibody/peptide-based probes is promising but mostly preclinical. It would be beneficial to add a paragraph discussing barriers to clinical translation, such as regulatory hurdles, production costs, or imaging system availability.
Response: We have added a summary of the key limitations of the included clinical trials of the NIR and antibody-based probes in the Conclusion section (page 12, lines 450-462).
“Fluorescently labeled monoclonal antibodies offer great potential for more personalized surgical strategies across a range of solid tumors, although significant challenges still need to be addressed. One key challenge is their requirement for preoperative administration several days before surgery, often necessitating additional hospital visits for patients [120]. Another drawback is the rapid quenching of fluorescence, which can compromise imaging accuracy during procedures [120]. Moreover, NIR dyes conjugated to monoclonal antibodies are still largely in the preclinical stage or are undergoing early-phase clinical trials involving a limited number of patients, which prevents any definitive conclusions regarding their clinical utility. Nonetheless, with continued advancements in imaging technologies, molecular targeting, and regulatory progress, NIR imaging holds a significant promise to become the gold standard for GBM surgery, potentially setting new benchmarks for surgical precision, tumor delineation, and overall treatment effectiveness.”
English revisions
Response: We have thoroughly revised the manuscript to improve clarity and ensure that all content is easily understood to the target readership.
Reviewer 2 Report
Comments and Suggestions for Authors
NICE AND INTRESTING WORK
ADD MORE REFERENCES
ADD MORE RECENT REFERENCES
MENTION THE INNOVATIONS OF THIS STUDY
ADD MORE DETAILED CONCLUSIONS-ADD MORE CONCLUSIONS
REMOVE THE TABLE 1 FROM THE CONCLUSIONS AND PUT IT INTO THE TEXT
ADD SOME MORE TABLES
ADD MORE ELEMENTS IN ALL THE SECTIONS OF THIS STUDY
Author Response
Thank you for taking the time to review our manuscript and for providing thoughtful and constructive feedback. We sincerely appreciate your comments, which have helped us improve the clarity and quality of our work.
Reviewer 2:
NICE AND INTRESTING WORK
ADD MORE REFERENCES
ADD MORE RECENT REFERENCES
MENTION THE INNOVATIONS OF THIS STUDY
ADD MORE DETAILED CONCLUSIONS-ADD MORE CONCLUSIONS
REMOVE THE TABLE 1 FROM THE CONCLUSIONS AND PUT IT INTO THE TEXT
ADD SOME MORE TABLES
ADD MORE ELEMENTS IN ALL THE SECTIONS OF THIS STUDY
Response: Thank you for your positive feedback and constructive suggestions. We have carefully revised the manuscript to address each of your points.
Reference List Update: We have revised the reference list to include several recent and relevant publications from the past 3–5 years. These updates enhance the background, support key findings, and better reflect current developments in the field such as references 124, 126, and 127.
Expanded Conclusion: The Conclusion section (pages 12–13) has been expanded to provide a more comprehensive summary of our findings, highlighting their implications, potential applications, and directions for future research.
Retention of Table 1: In response to Reviewer 3's suggestion, we have retained Table 1 as an informative summary. As noted by the reviewer, “The comparative table summarizing the different fluorescent agents is a valuable addition, providing a quick overview of their key characteristics.”
Addition of a New Table: We have added a new table to the manuscript (page 5) to further support the presented data.
Additional Content Added: We have included additional text in the following sections to enhance clarity and depth: FS section (page 8, lines 270–274), Antibody-Based Probes for Fluorescence Imaging (page 10, lines 345–349), Peptide-Based Probes for Fluorescence Imaging (page 11, lines 416–420), Conclusion section (pages 12–13).
Reviewer 3 Report
Comments and Suggestions for Authors
The review Nguyen and colleagues provides a comprehensive overview of fluorescence-guided surgery (FGS) techniques for glioblastoma (GBM) resection, covering established and emerging fluorophores. It discusses the mechanisms, clinical applications, limitations, and future directions of each technique. The review focuses on 5-aminolevulinic acid (5-ALA), fluorescein sodium (FS), indocyanine green (ICG), antibody-based probes, peptide-based probes, and the potential of near-infrared (NIR) imaging.
The authors should be commended for several key strengths:
Comprehensive Coverage: The review covers a wide range of FGS techniques, including established and emerging methods, providing a thorough overview of the field.
Clear and Concise: The writing is clear, concise, and easy to understand, making the information accessible to a broad audience.
Well-Structured: The review is logically organized, with distinct sections for each fluorophore and technique.
Detailed Explanations: The mechanisms of action and clinical applications of each fluorophore are explained in detail, providing valuable insights.
Balanced Perspective: The review presents a balanced perspective, discussing both the advantages and limitations of each technique.
Up-to-Date: The review includes recent research and clinical trials, reflecting the current state of the field.
Relevant References: The review cites a substantial number of relevant and recent publications, supporting the information presented.
Informative Table: The comparative table summarizing the different fluorescent agents is a valuable addition, providing a quick overview of their key characteristics.
However, some weaknesses were identified which, if addressed, will strengthen the review overall relevance and impact.
Limited Discussion of Cost-Effectiveness: While the review mentions the cost-effectiveness of 5-ALA, it lacks a detailed discussion of the cost-effectiveness of other FGS techniques, especially the newer, more advanced methods. This information is crucial for practical implementation and decision-making in clinical settings. A comparative cost analysis of the different fluorophores and imaging systems, if known, would strengthen the review.
Lack of Critical Analysis of Clinical Trials: The review summarizes several clinical trials but lacks a critical analysis of their methodologies and limitations. For instance, while mentioning sample sizes, it doesn't delve into the potential biases or confounding factors that might influence the results. A more critical appraisal of the clinical evidence would enhance the review's scientific rigor.
Insufficient Detail on Future Directions: While the review mentions future directions, such as the integration of NIR with complementary technologies and machine learning, these areas are not explored in sufficient detail. Expanding on these aspects, including specific examples of ongoing research and potential challenges, would provide a more complete picture of the field's trajectory.
Uneven Depth of Coverage: The review provides extensive detail on 5-ALA and FS but less on ICG and other emerging probes. While this might reflect the relative maturity of the different techniques, a more balanced discussion of the newer methods, including their potential advantages and limitations, would be beneficial. For example, the discussion of antibody-based probes could be expanded to include more details on their development, preclinical testing, and potential clinical applications.
Limited Discussion of Combination Modalities: The review briefly mentions the combination of FS and 5-ALA in one clinical trial but doesn't explore the potential of combining different FGS techniques or integrating them with other treatment modalities. A discussion of these possibilities, including potential synergistic effects and challenges, would add value to the review.
Priority Score: 85/100
Author Response
Thank you for taking the time to review our manuscript and for providing thoughtful and constructive feedback. We sincerely appreciate your comments, which have helped us improve the clarity and quality of our work.
Reviewer 3:
The review Nguyen and colleagues provides a comprehensive overview of fluorescence-guided surgery (FGS) techniques for glioblastoma (GBM) resection, covering established and emerging fluorophores. It discusses the mechanisms, clinical applications, limitations, and future directions of each technique. The review focuses on 5-aminolevulinic acid (5-ALA), fluorescein sodium (FS), indocyanine green (ICG), antibody-based probes, peptide-based probes, and the potential of near-infrared (NIR) imaging.
The authors should be commended for several key strengths:
Comprehensive Coverage: The review covers a wide range of FGS techniques, including established and emerging methods, providing a thorough overview of the field.
Clear and Concise: The writing is clear, concise, and easy to understand, making the information accessible to a broad audience.
Well-Structured: The review is logically organized, with distinct sections for each fluorophore and technique.
Detailed Explanations: The mechanisms of action and clinical applications of each fluorophore are explained in detail, providing valuable insights.
Balanced Perspective: The review presents a balanced perspective, discussing both the advantages and limitations of each technique.
Up-to-Date: The review includes recent research and clinical trials, reflecting the current state of the field.
Relevant References: The review cites a substantial number of relevant and recent publications, supporting the information presented.
Informative Table: The comparative table summarizing the different fluorescent agents is a valuable addition, providing a quick overview of their key characteristics.
However, some weaknesses were identified which, if addressed, will strengthen the review overall relevance and impact.
Limited Discussion of Cost-Effectiveness: While the review mentions the cost-effectiveness of 5-ALA, it lacks a detailed discussion of the cost-effectiveness of other FGS techniques, especially the newer, more advanced methods. This information is crucial for practical implementation and decision-making in clinical settings. A comparative cost analysis of the different fluorophores and imaging systems, if known, would strengthen the review.
Response: To date, no publication has provided a comprehensive comparison of the exact costs associated with the different methods such as microscopy, fluorescence detection, and user training. However, it is generally accepted that 5-ALA is the most expensive option, FS is the least expensive, and NIR-based systems fall in between. One study reported the cost per vial as follows: 5-ALA – $1,040.25; SF – $9.25; and NIR – $924.46 (PMID: 38818124).
Beyond cost-effectiveness, the safety profile of each method should also be taken into account when selecting the most appropriate technique. I have expanded the Discussion section (page 12, lines 438-440) to address these cost-effectiveness considerations in the context of fluorescence-guided surgery (FGS) for neurosurgery.
“In contrast, NIR fluorescence imaging offers several advantages, including deeper tissue penetration, reduced background autofluorescence, and real-time intraoperative visualization. These features allow for more accurate delineation of tumor margins, even in deep-seated or infiltrative lesions, thereby improving surgical precision and patient outcomes. Unlike 5-ALA, NIR agents are not metabolized and are exclusively cleared by the liver, with a short plasma half-life of approximately 3–4 minutes. In addition to its clinical benefits, NIR imaging may offer a more cost-effective option compared to 5-ALA, not only in terms of agent cost but also in handling and training requirements, while maintaining high imaging performance [37]. Given these benefits, NIR fluorescence imaging emerges as a promising alternative to traditional fluorophores like 5-ALA and FS, with the potential to become a new standard in glioma surgery (Table 1).”
Lack of Critical Analysis of Clinical Trials: The review summarizes several clinical trials but lacks a critical analysis of their methodologies and limitations. For instance, while mentioning sample sizes, it doesn't delve into the potential biases or confounding factors that might influence the results. A more critical appraisal of the clinical evidence would enhance the review's scientific rigor.
Response: We thank the reviewer for this valuable feedback. In response, we have added text discussing the limitations of FGS methods throughout the manuscript. These additions can be found in the FS section (page 8, lines 270–274), Antibody-Based Probes for Fluorescence Imaging (page 10, lines 345–349), Peptide-Based Probes for Fluorescence Imaging (page 11, lines 416–420), and the Conclusion section (pages 12–13).
Insufficient Detail on Future Directions: While the review mentions future directions, such as the integration of NIR with complementary technologies and machine learning, these areas are not explored in sufficient detail. Expanding on these aspects, including specific examples of ongoing research and potential challenges, would provide a more complete picture of the field's trajectory.
Response: We appreciate the reviewer’s suggestion to elaborate on the integration of NIR fluorescence imaging with complementary technologies and machine learning. We have expanded the discussion of future directions to include specific examples of current research efforts (page 12, line 466-489).
“The impact of NIR could be further enhanced by integrating it with complementary technologies such as handheld probes and intraoperative MRI (iMRI). These innovations improve detection sensitivity, compensate for intraoperative brain shift, and provide additional anatomical and functional information [20, 121-123]. For instance, the randomized clinical trial NCT01479686 in 188 high-grade gliomas demonstrated that iMRI significantly increased the extent of resection in glioma surgery, achieving a GTR rate of 83.85% compared to 50% with conventional neuronavigation [20]. Complementing this, a study evaluating the combined use of iMRI and 5-ALA found that in tumors near eloquent regions, 5-ALA alone yielded a 72% GTR rate, which rose to 100% when iMRI was added. Similarly, for tumors located within eloquent areas, GTR improved from 58% with 5-ALA alone to 71% with the integration of iMRI. Together, these findings underscore the potential of combining near-infrared (NIR) fluorescence imaging with iMRI to enhance tumor resection and ultimately improve patient outcomes [124].
Furthermore, the incorporation of machine learning (ML) algorithms can enhance specificity and predict tumor invasion patterns, contributing to the advancement of precision neurosurgery and image-guided therapy for GBM [125]. One study explored a fluorescence-based ML approach for intraoperative brain tumor classification by training and testing various ML models using fluorescence spectral data processed in Python. To optimize performance, the researchers employed strategies such as varying the number of pixels per sample, balancing class representation, and using 5-fold cross-validation for model tuning. Despite limitations stemming from noisy training data, tumor heterogeneity, and subjective margin labeling, the models achieved strong accuracy (up to 85.7%), with notable success in predicting IDH mutation status (93%). Future improvements, including the use of convolutional neural networks, enhanced spectral unmixing, and standardized labeling protocols, could further boost performance. Overall, this work supports the growing feasibility and potential of ML-integrated, fluorescence-guided systems to improve intraoperative decision-making and surgical outcomes in neurosurgery.”
Uneven Depth of Coverage: The review provides extensive detail on 5-ALA and FS but less on ICG and other emerging probes. While this might reflect the relative maturity of the different techniques, a more balanced discussion of the newer methods, including their potential advantages and limitations, would be beneficial. For example, the discussion of antibody-based probes could be expanded to include more details on their development, preclinical testing, and potential clinical applications.
Response: We thank the reviewer for this thoughtful comment. We have added a brief summary of the key limitations of the NIR antibody-based probes in the Conclusion section (page 12, lines 450-462).
“Fluorescently labeled monoclonal antibodies offer great potential for more personalized surgical strategies across a range of solid tumors, although significant challenges still need to be addressed. One key challenge is their requirement for preoperative administration several days before surgery, often necessitating additional hospital visits for patients [120]. Another drawback is the rapid quenching of fluorescence, which can compromise imaging accuracy during procedures [120]. Moreover, NIR dyes conjugated to monoclonal antibodies are still largely in the preclinical stage or are undergoing early-phase clinical trials involving a limited number of patients, which prevents any definitive conclusions regarding their clinical utility. Nonetheless, with continued advancements in imaging technologies, molecular targeting, and regulatory progress, NIR imaging holds a significant promise to become the gold standard for GBM surgery, potentially setting new benchmarks for surgical precision, tumor delineation, and overall treatment effectiveness.”
Limited Discussion of Combination Modalities: The review briefly mentions the combination of FS and 5-ALA in one clinical trial but doesn't explore the potential of combining different FGS techniques or integrating them with other treatment modalities. A discussion of these possibilities, including potential synergistic effects and challenges, would add value to the review.
Response: Thank you for the insightful suggestion. We have incorporated a discussion of the limitations of NIR, along with its potential synergistic effects and associated challenges (Page 12, lines 444-489).
“However, ICG’s strong affinity for plasma proteins can limit its accumulation at tumor sites, thereby reducing its specificity. To address this limitation, recent advancements in cancer imaging have focused on conjugating NIR dyes to monoclonal antibodies that selectively target tumor-associated antigens such as Cetuximab-IRDye 800, Bevacizumab-IRDye800CW, MCT4-ICG-NIR-II, and Miltuximab-NIR, as well as peptide-based probes such as BLZ-100. These targeted imaging agents have shown significant promise in improving intraoperative tumor visualization and guiding more effective resections. Fluorescently labeled monoclonal antibodies offer great potential for more personalized surgical strategies across a range of solid tumors, although significant challenges still need to be addressed. One key challenge is their requirement for preoperative administration several days before surgery, often necessitating additional hospital visits for patients [119]. Another drawback is the rapid quenching of fluorescence, which can compromise imaging accuracy during procedures [119]. Moreover, NIR dyes conjugated to monoclonal antibodies are still largely in the preclinical stage or are undergoing early-phase clinical trials involving a limited number of patients, which prevents any definitive conclusions regarding their clinical utility. Nonetheless, with continued advancements in imaging technologies, molecular targeting, and regulatory progress, NIR imaging holds a significant promise to become the gold standard for GBM surgery, potentially setting new benchmarks for surgical precision, tumor delineation, and overall treatment effectiveness.
The impact of NIR could be further enhanced by integrating it with complementary technologies such as handheld probes and intraoperative MRI (iMRI). These innovations improve detection sensitivity, compensate for intraoperative brain shift, and provide additional anatomical and functional information [20, 120-122]. For instance, the randomized clinical trial NCT01479686 in 188 high-grade gliomas demonstrated that iMRI significantly increased the extent of resection in glioma surgery, achieving a GTR rate of 83.85% compared to 50% with conventional neuronavigation [20]. Complementing this, a study evaluating the combined use of iMRI and 5-ALA found that in tumors near eloquent regions, 5-ALA alone yielded a 72% GTR rate, which rose to 100% when iMRI was added. Similarly, for tumors located within eloquent areas, GTR improved from 58% with 5-ALA alone to 71% with the integration of iMRI. Together, these findings underscore the potential of combining near-infrared (NIR) fluorescence imaging with iMRI to enhance tumor resection and ultimately improve patient outcomes [123].
Furthermore, the incorporation of machine learning (ML) algorithms can enhance specificity and predict tumor invasion patterns, contributing to the advancement of precision neurosurgery and image-guided therapy for GBM [124]. One study explored a fluorescence-based ML approach for intraoperative brain tumor classification by training and testing various ML models using fluorescence spectral data processed in Python. To optimize performance, the researchers employed strategies such as varying the number of pixels per sample, balancing class representation, and using 5-fold cross-validation for model tuning. Despite limitations stemming from noisy training data, tumor heterogeneity, and subjective margin labeling, the models achieved strong accuracy (up to 85.7%), with notable success in predicting IDH mutation status (93%). Future improvements, including the use of convolutional neural networks, enhanced spectral unmixing, and standardized labeling protocols, could further boost performance. Overall, this work supports the growing feasibility and potential of ML-integrated, fluorescence-guided systems to improve intraoperative decision-making and surgical outcomes in neurosurgery.”
Reviewer 4 Report
Comments and Suggestions for Authors
In this article, the authors present a review of fluorescence-guided techniques for gliomas. They summarize different techniques, their mechanisms, limitations, and clinical applications.
The review includes key information important for a comparative understanding of the topic.
I have one observation: on line 19, sodium fluorescein (SF) is mentioned, but on lines 21 and 61, sodium fluorescein (SF) is written.
I suggest including references from 2025 to update the information. For example:
Bailey D, Zacharia BE. Intraoperative imaging techniques to improve tumor detection in the surgical management of gliomas. Adv Cancer Res. 2025;166:103-135. doi: 10.1016/bs.acr.2025.05.001. Epub 2025 May 23. PMID: 40675684.
An W, Wang Z, Miao Q, Li Q. Organic probes for NO-activatable biomedical imaging: NIR fluorescence, self-luminescence, and photoacoustic imaging. Chem Sci. 2025 Jul 14. doi: 10.1039/d5sc03611a. Epub ahead of print. PMID: 40698164; PMCID: PMC12278105.
Elliot M, Ségaud S, Lavrador JP, Vergani F, Bhangoo R, Ashkan K, Xie Y, Stasiuk GJ, Vercauteren T, Shapey J. Fluorescence Guidance in Glioma Surgery: A Narrative Review of Current Evidence and the Drive Towards Objective Margin Differentiation. Cancers (Basel). 2025 Jun 17;17(12):2019. doi: 10.3390/cancers17122019. PMID: 40563668; PMCID: PMC12190578.
Rodriguez B, Brown CS, Colan JA, Zhang JY, Huq S, Rivera D, Young T, Williams T, Subramaniam V, Hadjipanayis C. Fluorescence-Guided Surgery for Gliomas: Past, Present, and Future. Cancers (Basel). 2025 May 30;17(11):1837. doi: 10.3390/cancers17111837. PMID: 40507320; PMCID: PMC12153724.
Author Response
Thank you for taking the time to review our manuscript and for providing thoughtful and constructive feedback. We sincerely appreciate your comments, which have helped us improve the clarity and quality of our work.
Reviewer 4:
In this article, the authors present a review of fluorescence-guided techniques for gliomas. They summarize different techniques, their mechanisms, limitations, and clinical applications.
The review includes key information important for a comparative understanding of the topic.
I have one observation: on line 19, sodium fluorescein (SF) is mentioned, but on lines 21 and 61, sodium fluorescein (SF) is written.
Response: We have updated line 21 to use only the abbreviation 'SF'. In line 61, we retain the full term 'sodium fluorescein (SF)' as it is the first mention in the Introduction.
I suggest including references from 2025 to update the information. For example:
Bailey D, Zacharia BE. Intraoperative imaging techniques to improve tumor detection in the surgical management of gliomas. Adv Cancer Res. 2025;166:103-135. doi: 10.1016/bs.acr.2025.05.001. Epub 2025 May 23. PMID: 40675684.
Response: We have now included this reference in the manuscript (Reference 124).
An W, Wang Z, Miao Q, Li Q. Organic probes for NO-activatable biomedical imaging: NIR fluorescence, self-luminescence, and photoacoustic imaging. Chem Sci. 2025 Jul 14. doi: 10.1039/d5sc03611a. Epub ahead of print. PMID: 40698164; PMCID: PMC12278105.
Response: This review article focuses extensively on organic probes for NO-activatable biomedical imaging, which falls outside the scope of my review; therefore, it was not included.
Elliot M, Ségaud S, Lavrador JP, Vergani F, Bhangoo R, Ashkan K, Xie Y, Stasiuk GJ, Vercauteren T, Shapey J. Fluorescence Guidance in Glioma Surgery: A Narrative Review of Current Evidence and the Drive Towards Objective Margin Differentiation. Cancers (Basel). 2025 Jun 17;17(12):2019. doi: 10.3390/cancers17122019. PMID: 40563668; PMCID: PMC12190578.
Response: We have now included this reference in the manuscript (Reference 126).
Rodriguez B, Brown CS, Colan JA, Zhang JY, Huq S, Rivera D, Young T, Williams T, Subramaniam V, Hadjipanayis C. Fluorescence-Guided Surgery for Gliomas: Past, Present, and Future. Cancers (Basel). 2025 May 30;17(11):1837. doi: 10.3390/cancers17111837. PMID: 40507320; PMCID: PMC12153724.
Response: We have now included this reference in the manuscript (Reference 127).